# A Comparative Study of Advanced Stationary Phases for Fast Liquid Chromatography Separation of Synthetic Food Colorants

**DOI:** 10.3390/molecules23123335

**Published:** 2018-12-15

**Authors:** Ivona Lhotská, Petr Solich, Dalibor Šatínský

**Affiliations:** Department of Analytical Chemistry, Faculty of Pharmacy in Hradec Králové, Charles University, Heyrovského 1203, Hradec Králové 50005, Czech Republic; lhotski@faf.cuni.cz (I.L.); solich@faf.cuni.cz (P.S.)

**Keywords:** monolithic column, porous shell column, food additive, dye, food colorant, chromatography, fast chromatography

## Abstract

Food analysis demands fast methods for routine control and high throughput of samples. Chromatographic separation enables simultaneous determination of numerous compounds in complex matrices, several approaches increasing separation efficiency and speed of analysis were involved. In this work, modern types of column with monolithic rod or superficially porous particles were employed and compared for determination of eight synthetic food dyes, their chromatographic performance was evaluated. During method optimization, cyano stationary phase Chromolith Performance CN 100 × 4.6 mm and Ascentis Express ES-CN 100 × 4.6 mm, 5 µm were selected for the separation of polar colorants. The separation was performed by gradient elution of acetonitrile/methanol and 2% water solution of ammonium acetate at flow rate 2.0 mL min^−1^. Mobile phase composition and the gradients were optimized in order to enable efficient separation on both columns. The method using fused-core particle column provided higher separation efficiency, narrow peaks of analytes resulted in increased peak capacity and shortening of analysis time. After the validation, the method was applied for analysis of coloured beers, soft drinks and candies.

## 1. Introduction

Food colorants are one of the food additives used extendedly, especially in beverages and candies. There are various natural colours, however their tint is dependent on pH and the colour can be lost during time and storage conditions [1]. Hence, more stable and cheaper artificial dyes obtained by chemical synthesis from highly purified oil products are used favourably.

Recently, there is a trend to return to natural origin of colorants in high-class products or intended for children consumption as the customers demand according to the latest findings [2]. Increased attention was paid to adverse effects of synthetic food colorants and their long-term toxicity. Several food dyes demonstrated positive results in genotoxicity tests [3,4] but usually the tested concentrations overtop levels present in food. Some azo dyes have already been banned due to their carcinogenic degradation products [5], azo dyes tartrazine and sunset yellow are suspected from xenoestrogenic behaviour [6]. Although remaining controversial, using of synthetic coloration seems to be safe, meeting the recommended acceptable daily intake (ADI) values [1], which are exposed to legislative limits [7]. However even with low consumption of colourings, hypersensitive reactions may occur in susceptible individuals. Many allergic and other immune reactive disorders have been reported [8,9,10]. Moreover, behavioural changes depending on diet are observed in sensitive children [9]. Ingestion of great amount of foreign antigenic load, such as sugar, synthetic food colorants and other additives has been connected with development of attention and hyperactivity disorders in children. Reasonably, the synthetic colours application and used concentration in food should be controlled. The food additives contained in particular foodstuff have to be listed on label and there are specified concentrations for each product [7].

There are plenty of possibilities for dyes determination as they have varying physico-chemical properties. Water-soluble food colorants such as azo-dyes, triarylmethane dyes, quinopthalone, or indigoid dyes differ in lipophilicity and pKa values. Oxidation of some dyes can be used for electrochemical determination [11]. The sensitive simultaneous determination of colorants has been achieved by capillary electrophoresis [12] and even miniaturized electrophoresis on a chip [13].

Spectrophotometry could be method of choice for analysing colourful substances. However, their simultaneous determination is very limited due to the similar spectral characteristics [14,15]. On the other hand, spectrophotometric detection is extensively utilized mainly in combination with chromatographic separation of analytes. After appropriate pre-extraction and clean-up, it is a very common and versatile method for simultaneous determination of numerous colourings in complex food matrices. Despite of some recent interesting applications [16], thin-layer chromatography has been almost superseded by liquid chromatography (HPLC-DAD) [17,18,19,20,21,22,23], similarly very fast and simple method of sequential injection chromatography has been reported [24]. Nowadays, in many cases mass spectrometry detection with mass confirmation has been involved [25,26]. Robust, sensitive and fast methods are required in food analysis. Fast chromatography methods represent current trend in increasing the speed of analyses. Several approaches are perspective, high temperature, sub-2 µm particles in ultra-pressurized systems and advanced technologies of stationary phases preparation such as superficially porous particles and monolithic column. Fast chromatography enables fastening of analysis without a loss of separation efficiency [27].

In this study, two methods using different technology of stationary phase for determination of food synthetic colours in drinks and candies were developed and validated. The chromatography columns packed with monolithic sorbent and superficially porous particles as the two important approaches in current trends in fast chromatography were chosen for comparison of their chromatographic performance. Those types of packing materials represent development in liquid chromatography providing higher separation efficiency and the speed of analysis [28].

## 2. Results and Discussion

### 2.1. Method Development

Most of the parameters were optimized on the monolithic column first. The middle polar cyano stationary phase was chosen as it showed good selectivity for mostly polar colorants. In advance, a mobile phase with high ionic strength containing ammonium acetate showed great influence of ‘salting out effect’ enabling increased interaction of compounds with stationary phase and thus good retention of the more polar colorants. The resolution of critical pairs of peaks (Tartrazine—Indigo carmine; Green S—Patent blue; Brilliant blue—Fast green) was finally solved with raising the concentration of ammonium acetate from 0.1% up to 2% and gradient elution program with methanol. Low back pressure due to lower mass transfer resistance of monolith enabled to use a flow rate 2 mL min^−1^.

Afterwards, fused-core particle column was taken of similar parameters (cyano stationary phase, 10 cm length and 4.6 mm diameter), the same buffer, flow rate and injection volume 10 µL. However, original gradient was optimized and it was taken advantage of acetonitrile in mobile phase for its lower viscosity, higher elution power and better resolution of peaks on this stationary phase.

### 2.2. Method Validation and Comparison

Two methods for determination of food synthetic colours employing either monolithic sorbent or superficially porous particles columns were developed, their chromatographic performances were evaluated. Comparison was performed in terms of repeatability, peak symmetry, peak capacity and linear range (column loading capacity).

Peak capacity representing separation efficiency parameter is calculated from peak width. As can be seen in Figure 1 and Figure 2 and according the values in Table 1, fused-core column in combination with acetonitrile in mobile phase provided very narrow peaks, in shorter gradient time, leading to higher peak capacity. The analytes were concentrated in narrow zones represented by narrow peaks with higher response. That lowered limit of detection to 0.2 mg L^−1^; but evidentially the detector was overloaded soon (upper limit of detection 100 mg L^−1^), although in case of higher concentrations, simple dilution of sample would solve that.

On the monolithic column, the resolution of all colorants was achieved only with longer methanol gradient. Probably worsened by the solvent used, the peaks were not so narrow, high and symmetric. As is mentioned in Table 2, the linear range is than shifted to higher concentrations (0.5–200 mg L^−1^). In addition, the compounds order change was observed (retention times in Table 1). It was caused by using acetonitrile instead of methanol and better solubility of Green S in acetonitrile mobile phase used for fused-core particle column. The separation on monolithic column showed to be more complicated due to a strong peak tailing of Patent blue. This negative effect was not observed on fused core particle column. Moreover, three critical pairs Tartrazine—Indigo carmine; Green S—Patent blue; Brilliant blue—Fast green, depicted in Figure 2 showed peak resolution factor lower than 1.5 which complicated reliable validation of the method. The peaks of Tartrazine and Indigo carmine were not well separated on monolithic column and their quantification was possible only due to a different spectral characteristics. On the other hand, the advantages of the monolithic column method were lower back pressure and less interferences on blank chromatogram. Repeatability of both methods (RSD in (%)), 0.1–4.5% for monolithic column and 0.9–3.7% for fused-core column and correlation coefficients of regression equations were similar. Details are mentioned in Table 2.

### 2.3. Analysis of Real Samples

After previous evaluation, the method showing efficient and fast separation on fused-core column was chosen for application on real samples analysis. Soft drinks (including syrups and powders for syrup preparation), candies and seasonal Easter green beers were tested. Liquid samples were just filtered and 10 µL of sample solution was directly injected. Syrups and powders were diluted and candies extracted in mixture of MeOH and water 50:50 (*v*/*v*). Samples were filtered before analysis through PTFE filter, nylon filters were excluded due to partial adsorption of analytes.

Identification of colorants in samples was confirmed by comparison of peak spectra with estimated colour standard solution spectra, details can be seen in Appendix A. All detected synthetic colorants were enlisted in the product label as legislative demands; their concentrations were in reasonable limits. Sample chromatograms of coloured drinks and candies are shown in Figure 3 and Figure 4.

## 3. Materials and Methods

### 3.1. Chemicals and Materials

Standards of synthetic colorants Quinoline yellow (95%), Tartrazine (99%), Sunset yellow (95%), Fast green FCF (98%), Green S (98%), Indigo carmine (98%) and Patent blue (97%) were purchased from Fluka (Neu-Ulm, Germany). Brilliant blue FCF was supplied by Sigma-Aldrich (Prague, Czech Republic). Glacial acetic acid was produced by Fluka, ammonium hydroxide 26% and HPLC-grade solvents acetonitrile and methanol were obtained from Sigma-Aldrich. The ultra-pure water purification for mobile phase preparation was carried out through a Milli-Q (Millipore, Bedford, MA, USA). Other chemicals and used materials were of analytical grade. Chromatographic column Chromolith Performance CN 100 × 4.6 mm with guard column Chromolith CN 5 × 4.6 mm was ordered from Merck (Prague, Czech Republic) and Ascentis Express ES-CN 100 × 4.6 mm, 5 µm from Sigma-Aldrich (Supelco).

Seasonal beers on tap were collected from local bars and just filtered to vial (0.45 µm PTFE). Other samples of drinks and candies were bought in local shop. Soft drinks were prepared by dissolving of powder or syrup in water at first, keeping the manufacturer instructions. Candies were dissolved in mixture of MeOH and water 50:50 (*v*/*v*) using ultrasound. Candy extracts were five times diluted before HPLC analysis due to high viscosity. All samples were filtered before analysis.

### 3.2. Chromatographic System

Chromatographic system Shimadzu Prominence consisted of a SIL-20AC autosampler, LC-20AD solvent delivery modules with a DGU-AS on-line degasser, a CTO-20AC column oven, an SPD-M20A (DAD) detector and a CBM-20A communication module, all HPLC system was from Shimadzu Corporation (Kyoto, Japan). The system control, data acquisition and data evaluation were performed by Shimadzu “LC Lab-Solution” software (Shimadzu Corporation).

### 3.3. Preparation of Standard Solutions

Standard stock solutions of individual colorants in concentration 4000 mg L^−1^ were prepared, 5 mg of Green S, Brilliant blue FCF, Patent blue, Fast green FCF were dissolved in methanol, Sunset yellow and Indigo carmine in water, Quinoline yellow in tetrahydrofuran and Tartrazine in methanol-water mixture (4:1, (*v*/*v*)). The standard stock solutions were stored in freezer. The mixed standard working solution in concentration 500 mg L^−1^ of each dye was prepared by mixing aliquot amount of all eight standard solutions. The calibration standard solutions were dissolved in the mixture of acetonitrile with water (1:1) over the concentration range from 0.2 mg L^−1^ to 200 mg L^−1^.

### 3.4. Method I: Chromatographic Separation on Monolithic Column

In the first method, eight synthetic food colorants of varying physico-chemical properties were separated on monolithic column Chromolith Performance CN 100 × 4.6 mm with guard column Chromolith CN 5 × 4.6 mm by gradient elution, tempered at 30 °C. The separation was initiated with 100% water solution of 2% acetic acid adjusted at pH 7.0 with ammonia (solvent B) at a flow rate of 2.0 mL min^−1^ for 2 min, followed by linear gradient of methanol (solvent A) to 40% in 10th min. Total analysis last 12 min, including 2 min of equilibrating with initial conditions after separation. Injection volume was 10 µL. Colorants were detected according to their absorption maximum at wavelengths 625 nm for the blue colours (Indigo carmine, Brilliant blue FCF, Paten blue, Fast green FCF and Green S), 482 nm for Sunset yellow and 420 nm for the rest of yellow colours (Quinoline yellow, Tartrazine).

### 3.5. Method II: Chromatographic Separation on Fused-Core Particle Column

For the second method, Ascentis Express ES-CN 100 × 4.6 mm, 5 µm particle size column with superficially porous particles was utilized for separation. The buffer (solvent B) remained water solution of 2% acetic acid adjusted at pH 7.0 with addition of ammonia; organic part of the mobile phase was acetonitrile (solvent A). A gradient elution started after conditioning with water mobile phase by raising solvent A to 20% in 1.5 min, the ratio was held to 2nd min, then slowly increased to 25% in 3.5 min and finally to the 50% in 5th min. The system was equilibrated back to 100% of solvent B for 2 min. Total analysis time for separation of eight colorants was 7 min. All the other conditions were kept the same for both methods.

## 4. Conclusions

Two methods for determination of synthetic food colorants using modern stationary phases were developed with different optimized mobile phase composition and gradient elution to achieve efficient separation of all eight colorants. Applying fast chromatography approach of advanced technology of column packing material provided reduction of back pressure and fastening the separation. We achieved significant shortening of analysis of eight colorants in comparison with previously published methods [17,18,19,20,23]. One work published separation of eight colorants using core-shell column in time less than six min [29]. However, our work based on using cyano stationary phase showed lower retention of colorants and lower back pressure compared with usual C-18 stationary phases. This fact has not been published yet.

Especially, the method using combination of column with superficially porous particles and acetonitrile in mobile phase showed good separation efficiency with higher peak capacity and better peak symmetry. All compounds were separated to baseline in only 7 min. Their peaks were symmetric, very narrow and with high detector response; hence the linear range was limited and the detector was overloaded in concentration higher than 100 mg L^−1^. Otherwise, broadening of peaks on monolithic column was observed, probably due to greater dispersion of zones on mesopores of monolithic sorbent. Both columns with cyano stationary phase exhibited extremely low back pressure and higher flow rates accelerating analysis were enabled.

## Figures and Tables

**Figure 1 molecules-23-03335-f001:**
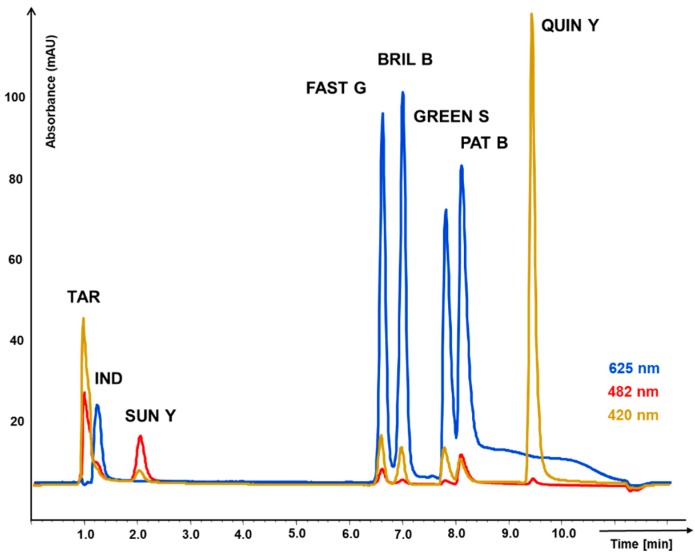
Chromatogram of 8 food synthetic dyes separation on monolithic column Chromolith Performance CN 100 × 4.6 mm with guard column Chromolith CN 5 × 4.6 mm. Gradient elution with mobile phase comprising of (A) methanol and (B) water solution of 2% acetic acid (pH 7.0). Detection of individual dyes using the wavelength according to their maximal absorption.

**Figure 2 molecules-23-03335-f002:**
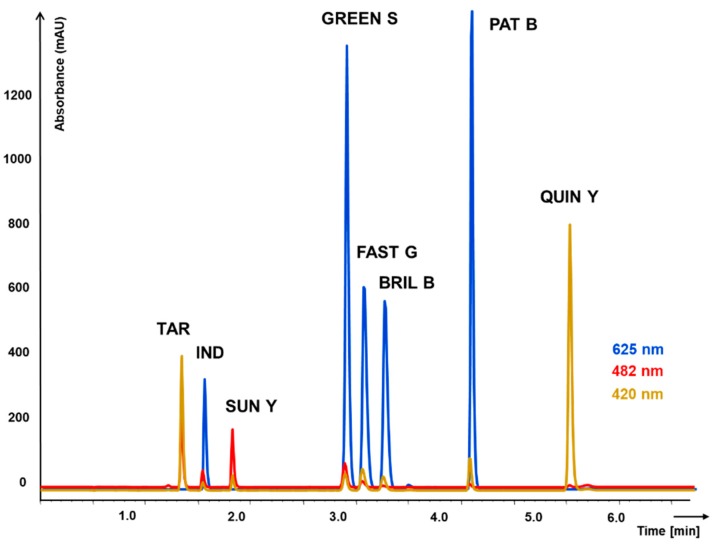
Chromatogram of 8 food synthetic dyes separation on fused-core column Ascentis Express ES-CN 100 × 4.6 mm, 5 µm. Gradient elution with mobile phase comprising of (A) acetonitrile and (B) water solution of 2% acetic acid (pH 7.0). Detection of individual dyes using the wavelength according to their maximal absorption.

**Figure 3 molecules-23-03335-f003:**
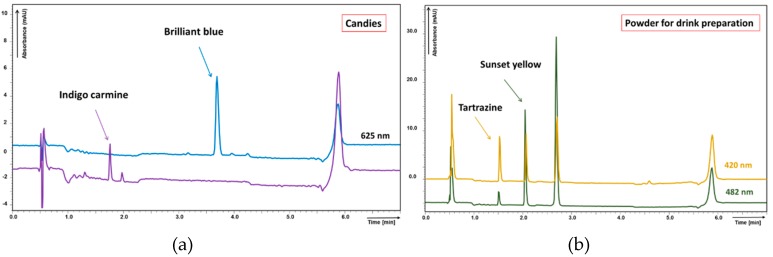
Chromatograms of sample matrices (analysed by method II): (**a**) candies and (**b**) powder for drink preparation. Found colorants in concentration: (**a**) indigo carmine 0.62 mg L^−1^, brilliant blue 1.29 mg L^−1^; (**b**) tartrazine 9.75 mg L^−1^, sunset yellow 71.38 mg L^−1^.

**Figure 4 molecules-23-03335-f004:**
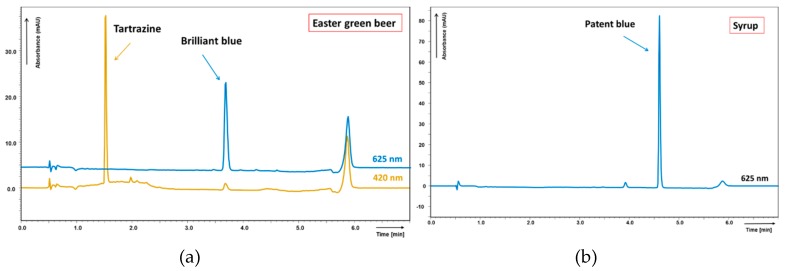
Chromatograms of sample matrices (analysed by method II): (**a**) seasonal coloured beer, (**b**) syrup. Found colorants in concentration: (**a**) tartrazine 3.49 mg L^−1^, brilliant blue 2.18 mg L^−1^; (**b**) patent blue 44.51 mg L^−1^.

**Table 1 molecules-23-03335-t001:** System suitability test results.

Analyte	λ (nm)	t_R_ (min) ^a^	Repeatability		Peak Symmetry	w_h_ ^b^	P_c_ ^c^
			t_R,_ RSD (%) ^d^	Peak Area, RSD (%) ^e^			
**Monolithic Column**							
Tartrazine	420	0.95	0.3	0.4; 0.1; 0.9	2.47	0.12	22.37
Indigo carmine	625	1.18	0.3	1.9; 0.4; 1.1	1.38	0.12	21.83
Sunset yellow	482	1.99	0.2	0.4; 0.1; 0.8	1.32	0.12	21.49
Fast green	625	6.53	0.1	0.3; 0.6; 0.7	1.19	0.12	22.37
Brilliant blue	625	6.91	0.1	0.3; 0.4; 0.5	1.09	0.12	21.66
Green S	625	7.67	0.1	1.6; 0.9; 1.0	1.53	0.16	16.72
Patent blue	625	7.96	0.1	0.7; 2.1; 0.6	2.32	0.18	15.12
Quinoline yellow	420	9.44	0.1	1.3; 0.4; 4.5	1.29	0.11	22.93
**Fused-core Column**							
Tartrazine	420	1.51	0.2	1.5; 1.8; 1.0	1.60	0.02	60.38
Indigo carmine	625	1.74	0.2	1.7; 1.8; 1.5	1.82	0.03	48.50
Sunset yellow	482	2.05	0.1	1.6; 1.8; 1.3	1.77	0.03	50.14
Green S	625	3.26	0.1	3.7; 1.8; 1.1	1.33	0.05	29.50
Fast green	625	3.45	0.1	1.9; 2.0; 1.5	1.53	0.04	34.14
Brilliant blue	625	3.67	0.1	1.8; 1.8; 1.2	1.45	0.05	31.98
Patent blue	625	4.60	0.1	2.1; 2.0; 1.2	1.71	0.04	41.71
Quinoline yellow	420	5.67	0.1	1.8; 1.5; 0.9	1.38	0.04	39.51

^a^ Retention time; ^b^ Peak width at 50% of the peak height; ^c^ Peak capacity (counting with time of gradient elution without final equilibration, 10 min and 5.7 min for monolithic and fused-core column, respectively); ^d^ RSD was calculated from eight injections of standard mixture at concentration level c = 50 mg L^−1^; ^e^ RSD was calculated from six injections of standard mixture at concentration levels: c_1_ = 200 mg L^−1^, c_2_ = 50 mg L^−1^, c_3_ = 5 mg L^−1^ on monolithic column and c_1_ = 100 mg L^−1^, c_2_ = 50 mg L^−1^, c_3_ = 5 mg L^−1^ on fused-core column.

**Table 2 molecules-23-03335-t002:** The analytical characteristics of the developed method in comparison of two types of columns.

Analyte	Linear Range (mg L^−1^) ^a^	Regression Equation	R^2^	LOD (mg L^−1^)	LOQ (mg L^−1^)
**Monolith**					
Tartrazine	0.5–200	y = 11,250x + 21,652	0.9973	0.03	0.09
Quinoline yellow	0.5–200	y = 20,143x + 106,689	0.9934	0.03	0.11
Sunset yellow	2–200	y = 2149x + 7683	0.9967	0.26	0.86
Indigo carmine	0.5–200	y = 6834x + 23,541	0.9945	0.05	0.17
Fast green	0.5–200	y = 28,509x + 96,884	0.9958	0.02	0.08
Brilliant blue	0.5–200	y = 31,272x + 116,370	0.9954	0.02	0.06
Green S	0.5–200	y = 23,582x + 89,043	0.9961	0.03	0.10
Patent blue	0.5–200	y = 39,910x − 16,514	0.9997	0.02	0.06
**Fused-Core**					
Tartrazine	0.2–100	y = 12,770x − 3835	0.9999	0.05	0.17
Quinoline yellow	0.2–100	y = 30,986x − 13,046	0.9999	0.05	0.17
Sunset yellow	0.5–100	y = 2042x − 42	0.9999	0.20	0.67
Indigo carmine	0.5–100	y = 11,438x − 6214	0.9999	0.08	0.25
Green S	0.2–100	y = 45,731x − 78,955	0.9955	0.02	0.05
Fast green	0.2–100	y = 34,452x − 23,308	0.9998	0.02	0.06
Brilliant blue	0.2–100	y = 33,215x − 22,166	0.9998	0.02	0.07
Patent blue	0.2–100	y = 47,507x − 145,350	0.9908	0.01	0.04

^a^ Standard solutions at 0.2 mg L^−1^, 0.5 mg L^−1^, 2 mg L^−1^, 5 mg L^−1^, 10 mg L^−1^, 25 mg L^−1^, 50 mg L^−1^, 75 mg L^−1^, 100 mg L^−1^, 200 mg L^−1^ concentration level were measured, each in triplicate.

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
