# Peer review of "A Comparative Study of Advanced Stationary Phases for Fast Liquid Chromatography Separation of Synthetic Food Colorants"

_molecules, 2018, doi:10.3390/molecules23123335_

Reviewer 1 Report

In my opinion, in the present form the manuscript (molecules-395074) entitled ‘A comparative study of modern stationary phases for fast liquid chromatography separation of synthetic food colorants’ described by Ivona Lhotská, Petr Solich and Dalibor Šatínský can be recommended for publication in the Molecules after major revision.

 My remarks and recommendations to the Molecules manuscript are as follows:

 Page 5, Analysis of real samples:

Identification of the components of the samples by comparison of their retention factors alone is decisively not sufficient owing to similar retention of various accompanying compounds in the sample. Therefore, it is necessary to confirm their presence in the sample by comparison of the spectra of the analytes with those in spectra libraries (and (if possible) determination of the purity peaks).

Authors do not use the DAD detector. Therefore Authors, can obtain a spectrum "at stopped flow" for a specific retention time of the analyte.

If possible, also, the matrix effect may be evaluate and add in the text.

 Authors should add:

-        comparison of the spectra of the analytes with those in spectra libraries,

-        authors may evaluate matrix effect and influence of them on the results,

-        authors should add values of CCα and CCβ,

-        authors should add a comparison of values of N/m for both column for: monolithic column Chromolith Performance CN and fused-core column Ascentis Express ES-CN, and add what are a peak performances measured on different columns (number of theoretical plates, etc.)?,

-        what is a value of mean square errors (MSE)?

The work is well done, but Authors should emphasize the novelty of this paper.

In general, the conclusion should be revise and partially rewritten.

Please indicate clearly what is new with your manuscript (Conclusions) for the Molecules, especially in comparison to earlier of publication(s).

Author Response

Dear Editor,

 Enclosed you can find the revisions and detailed responses to reviewers concerning the manuscript ID: molecules-395074 Title: A comparative study of modern stationary phases for fast liquid chromatography separation of synthetic food colorants; Corresponding author : Dr. Dalibor Å atínský

In the revisions, the comments of the reviewers have been taken into consideration. We prepared a revised version of our manuscript. Further details (according the reviewers comments) for increasing the relevance of the reported results were discussed and incorporated to the revised text. We hope that the required changes significantly raised the quality of the paper and satisfied the reviewer requirements.

Our responses (blue highlighted) to the following comments:

Reviewer 1:

My remarks and recommendations to the Molecules manuscript are as follows:

Page 5, Analysis of real samples:

Identification of the components of the samples by comparison of their retention factors alone is decisively not sufficient owing to similar retention of various accompanying compounds in the sample. Therefore, it is necessary to confirm their presence in the sample by comparison of the spectra of the analytes with those in spectra libraries (and (if possible) determination of the purity peaks).

All details are mentioned bellow in answers. Comparison of spectral data of samples and standards was added in supplementary material.

Authors do not use the DAD detector. Therefore Authors, can obtain a spectrum "at stopped flow" for a specific retention time of the analyte.

We did use DAD detector. DAD is basic equipment of our HPLC system. We can check all spectral data during and after analysis. This information was added to specification of HPLC instrument.

If possible, also, the matrix effect may be evaluate and add in the text.

In our study, no matrix effects were observed. We check the spectral data and peak purity during the method validation.

 Authors should add:

-        comparison of the spectra of the analytes with those in spectra libraries,

Very good remark, thank you. Of course, we confirmed identity of peaks by comparison with standard solution spectra. We added this missing information to text and as you recommended, supplementary figures containing those spectra are provided.

-        authors may evaluate matrix effect and influence of them on the results,

Matrix effect evaluation is necessary in case of mass spectrometry detection when affecting ionization and correct quantification of analyte in matrix. In case of spectrophotometric detection, matrix effects are not used to be specified. Peak purity and spectral data were checked without negative results.  Potential negative influence in our method was eliminated by matrix matched calibration and their minor effect has been observed in previous study already. For further details, you can see the article: Stachová et al. (2016): Determination of green, blue and yellow artificial food colorants and their abuse in herb-coloured green Easter beers on tap. Food Addit. Contam.Part A, 33, 1139-1146.

-        authors should add values of CCα and CCβ,

Instead of calculated values of detection capability (CCβ) and decision limit (CCα), we preferred to estimate such parameters experimentally by dilution of fortified blank matrix, as limit of detection and limit of quantification (enlisted in Table 2). More precise and statistically correct CCα and CCβ values shall be encountered in case of trace analyses, determination of residual compounds or toxic contaminants. In case of food colorants present in higher concentrations they are not crucial. In contrary, experimentally confirmed LOQ values might be more predicative.

-        authors should add a comparison of values of N/m for both column for: monolithic column Chromolith Performance CN and fused-core column Ascentis Express ES-CN, and add what are a peak performances measured on different columns (number of theoretical plates, etc.)?,

Chromatographic performance was evaluated as peak capacity, which is more adequate in case of gradient elution. Presenting number of theoretical plates in case of gradient elution leads to misleading data. This is basic fact in chromatography. Therefore, we used well established peak capacity. The results are listed in Table 1. We emphasized this fact in 2.2. Method validation section (line 93).

-        what is a value of mean square errors (MSE)?

The mean square errors are expressed as repeatability in form of RSD values in (%), enlisted in Table 1, ranging 0.1-4.5% for monolithic column and 0.9-3.7% for fused core column. We added this information also to text.

The work is well done, but Authors should emphasize the novelty of this paper.

In general, the conclusion should be revise and partially rewritten.

Please indicate clearly what is new with your manuscript (Conclusions) for the Molecules, especially in comparison to earlier of publication(s).

Conclusion was enriched by novelty statement in comparison with earlier publications and partially rewritten.

Reviewer 2 Report

Cooments;

Please rephrase ( for better understanding) the sentence Line 29- Line 31. 

Please provide the numeric scale ( absorbance) for Figure 1 and Figure 2. 

Can the authors provide also the quantities of the colorants identified in Figure 3?

I suggest to the authors to integrate more data about fast chromatography!

Author Response

Reviewer 2:

Comments and Suggestions for Authors

Cooments;

Please rephrase ( for better understanding) the sentence Line 29- Line 31. 

As recommended, this sentence was rewritten and simplified.

Please provide the numeric scale ( absorbance) for Figure 1 and Figure 2. 

The purpose of these graphics was to show peak shapes and resolution so we did not considered it necessary. Nevertheless, the numeric scale is provided in updated figures.

Can the authors provide also the quantities of the colorants identified in Figure 3?

The found concentrations are now described in figure captions (Figure 3 and Figure 4).

I suggest to the authors to integrate more data about fast chromatography!

New paragraph introducing to fast chromatography approaches, including new reference of review about fast chromatography in food analysis, was added at the ending of introduction section.

Reviewer 3 Report

The proposed article is presented in a nice way.

1)     Line 68: Please put material & methods part to section 2 and respectively results and discussion part to section 3. It is more usual in this way.  

2)     Line 145: Please enlarge the 4 chromatograms. It would be better to put each one down the other.

3)     Why you did not optimize the parameters on the fused-core particle column as this was the proposed column?

4)     It would be nice to compare your results with other methods concerning food colorants.

5)     Also you could give more detail about the parameters which were optimized and the obtained results

Author Response

Reviewer 3:

1)     Line 68: Please put material & methods part to section 2 and respectively results and discussion part to section 3. It is more usual in this way.  

This is required Molecules’ layout, please see Instruction for Authors and other articles. It cannot be done.

2)     Line 145: Please enlarge the 4 chromatograms. It would be better to put each one down the other.

Those 4 chromatograms are now divided into 2 pictures (Figure 3 and new Figure 4) and enlarged (also the text inside) for better visibility.

3)     Why you did not optimize the parameters on the fused-core particle column as this was the proposed column?

The reason is jut time consecution. At first, we developed a method for colorants determination using monolithic column. In that time, we also optimized composition of mobile phase and studied salting out effect on the colorants retention, published in (Stachová et al. (2016): Determination of green, blue and yellow artificial food colorants and their abuse in herb-coloured green Easter beers on tap. Food Addit. Contam.Part A, 33, 1139-1146.). Afterwards, we wanted to compare separation that can be achieved on monolithic or porous shell particles column with the same chemistry and optimized the second method to achieve more effective and faster separation.

4)     It would be nice to compare your results with other methods concerning food colorants.

Comparison with other methods mentioned in introduction was added to conclusion section. Our separation on fused-core column was very fast in comparison with previously published LC separations.

5)     Also you could give more detail about the parameters which were optimized and the obtained results

The best resolution in the shortest time was the most important parameter and obtained results of both methods are presented. Methods were optimized in similar way for the reason of comparison, it means the same column parameters, stationary phase chemistry, mobile phase, but optimized gradient elution. All parameters of validation used for comparison are already presented in the paper.

Round  2

Reviewer 1 Report

In my opinion, in the present form the manuscript (molecules-395074) entitled ‘A comparative study of advanced stationary phases for fast liquid chromatography separation of synthetic food colorants’ described by Ivona Lhotská, Petr Solich and Dalibor Å atínský can be recommended for publication in the Molecules.

Reviewer 2 Report

changes accepted!

Reviewer 3 Report

Figure 3 and 4 should be enlarged more.